# Individual Cerebral Blood Flow Responses to Transcranial Direct Current Stimulation at Various Intensities

**DOI:** 10.3390/brainsci10110855

**Published:** 2020-11-13

**Authors:** Craig D. Workman, Alexandra C. Fietsam, Laura L. Boles Ponto, John Kamholz, Thorsten Rudroff

**Affiliations:** 1Department of Health and Human Physiology, University of Iowa, Iowa City, IA 52242, USA; craig-workman@uiowa.edu (C.D.W.); alexandra-fietsam@uiowa.edu (A.C.F.); 2Department of Radiology, University of Iowa Hospitals and Clinics, Iowa City, IA 52242, USA; laura-ponto@uiowa.edu; 3Department of Neurology, University of Iowa Hospitals and Clinics, Iowa City, IA 52242, USA; john-kamholz@uiowa.edu

**Keywords:** tDCS, neuroimaging, positron emission tomography, cerebral blood flow, multiple sclerosis

## Abstract

Transcranial direct current stimulation (tDCS) has been shown to alter cortical excitability. However, it is increasingly accepted that tDCS has high inter- and intra-subject response variability, which currently limits broad application and has prompted some to doubt if the current can reach the brain. This study reports individual cerebral blood flow responses in people with multiple sclerosis and neurologically healthy subjects that experienced 5 min of anodal tDCS at 1 mA, 2 mA, 3 mA, and 4 mA over either the dorsolateral prefrontal cortex (DLPFC) or the primary motor cortex (M1). The most notable results indicated anticipated changes in regional cerebral blood flow (rCBF) in two regions of one DLPFC subject (2 mA condition), and expected changes in one M1 subject in the 2 mA and 4 mA conditions and in another M1 subject in the 2 mA condition. There were also changes contrary to the expected direction in one DLPFC subject and in two M1 subjects. These data suggest the effects of tDCS might be site-specific and highlight the high variability and individualized responses increasingly reported in tDCS literature. Future studies should use longer stimulation durations and image at various time points after stimulation cessation when exploring the effects of tDCS on cerebral blood flow (CBF).

## 1. Introduction

Multiple sclerosis (MS) is a debilitating central nervous system disease that affects ~2.3 million people worldwide [1]. MS is characterized by disabling refractory symptoms (e.g., neuropathic pain, gait disturbance) [2,3] that significantly reduce quality of life [4]. Therefore, potential therapies, such as transcranial direct current stimulation (tDCS), that might address these symptoms represent a much-needed accessory to existing treatment modalities. tDCS involves the application of weak electrical currents that penetrate the scalp and are purported to alter the cortical excitability of the underlying brain areas, with the anode increasing excitability and the cathode decreasing excitability (i.e., inhibition) [5,6]. This neuromodulation technique has several desirable characteristics, including ease of use, low cost, and potential for in-home use [7]. However, as more results from tDCS studies continue to be published, it is becoming increasingly evident that this neuromodulatory tool is hampered by high inter- and intra-subject variability [8,9,10]. Indeed, the results of some meta-analyses challenge the efficacy of tDCS [11,12,13], although a group of tDCS experts has argued against some of these null meta-findings [14]. Nevertheless, there is a growing debate about the overall value of tDCS. Additionally, because ~50% of subjects lack a response to tDCS or experience unanticipated excitability changes (e.g., excitation under the cathode) [15,16], individualized analyses to identify tDCS responders and non-responders might represent a more efficient means of determining stimulation efficacy than group-wise comparisons.

In addition to high variability, a recent article questioned if the commonly used tDCS intensities (≤2 mA) can exceed scalp/skull shunting with enough residual current to reliably influence cortical excitability [17]. Similarly, a transcranial alternating current stimulation (tACS) study suggested that stimulation effects might stem, at least in part, from peripheral nerve activation and not from direct cortical entrainment [18]. Therefore, continued mechanistic investigations of the cortical effects of brain stimulation are required. One possible modality to investigate these mechanisms is positron emission tomography (PET). PET leverages the use of various radiotracers to quantify biological and neurological processes [19]. For example, the radiotracer [^15^O]water provides a measure of cerebral blood flow (CBF). Because of a strong association with glucose metabolism (i.e., coupling between metabolism and blood flow), CBF represents an indirect measure of brain function [20]. [^15^O]water-PET imaging has been used to evaluate cerebral activity [21] and changes in CBF from tDCS [22] in people with MS (PwMS).

Our lab previously presented a pilot analysis of the effects of dorsolateral prefrontal cortex (DLPFC) tDCS on CBF in three PwMS [22]. We found no immediate group-wise effects from 5 min of tDCS at sham, 1 mA, 2 mA, 3 mA, or 4 mA intensities. This study was designed to mimic the parameters used by Nitsche and Paulus [5,6], who found significant excitability increases/decreases from 5 min of 1 mA tDCS using transcranial magnetic stimulation (TMS) motor evoked potential (MEP). However, potentially important differences between our pilot study and those of Nitsche and Paulus were the study population, stimulation location, and the timing of the imaging/excitability determination. Specifically, Nitsche and Paulus [5,6] stimulated the primary motor cortex (M1) of healthy adults and performed their first excitability assessment 1 min after the cessation of stimulation, while our study [22] stimulated the DLPFC (selected to prevent task-related and spontaneous movement from influencing blood flow changes) of PwMS and imaged *immediately* after tDCS ended. Therefore, given the data suggesting high variability/individualized responses to tDCS (above), one purpose of this study was to explore the unique CBF responses to tDCS of the three MS subjects from Workman, et al. [22] (Experiment 1). The second purpose was to present individual responses of the effects of tDCS from a follow-up pilot investigation of the short-term effects (1 min post-tDCS) of stimulation over M1 in three neurologically healthy adults (Experiment 2) and to compare and contrast these findings with Experiment 1. Based on the results of the group-wise analysis [22] and the slight alterations to the protocol, few individual blood flow changes from DLPFC tDCS, while individual blood flow changes from M1 tDCS, potentially in a dose dependent manner, were expected.

## 2. Materials and Methods

The effects of tDCS on CBF were investigated with semi-quantitative [^15^O]water-PET, measurements [23,24,25,26,27]. A sample of three people with relapsing-remitting MS were recruited for Experiment 1, and three neurologically healthy men were recruited for Experiment 2 (Table 1). Two blocks of six PET scans were completed. Each block contained one baseline scan and five scans following randomized tDCS intensities (sham, 1, 2, 3, and 4 mA). The order of tDCS intensities was re-randomized during the second block, which was performed to verify the reliability of the first block. Therefore, each subject underwent a single session with 12 total scans. Figure 1 displays the stimulation and scan protocol. In Experiment 1, the subjects performed a counting task (1, 2, 3, 1, 2, 3, …) at a pace of one per second for the duration of each scan (100 s). For Experiment 2, the subjects completed a handgrip task by opening and closing their hand at a pace of one complete open/close cycle every two seconds during scanning. In both experiments, the subjects were audibly and visually monitored to ensure they performed the task as directed. The purpose of performing the counting and hand opening/closings tasks was to avoid diverse brain activity differences from various “resting” brain states between scans [28]. Thus, these tasks were simple, relevant to the brain areas stimulated, and helped occupy the subjects’ attention to facilitate similar baseline activity levels in all scans. The protocol was approved by the University of Iowa Institutional Review Board, the study was performed in accordance with the Declaration of Helsinki, and all subjects provided written informed consent before participation (clinicaltrials.gov NCT04033133).

Stimulation was administered with a battery-operated tDCS device (Soterix Medical Inc., New York, NY, USA). It has previously been shown that 5 min of tDCS over M1 was sufficient to elicit a significant increase in MEP amplitude and that the effects of stimulation returned to baseline after 5 min and were completely dissipated 10 min post-stimulation [6]. Moreover, another study demonstrated that 4 min of motor cortex stimulation induced significant changes in blood flow [29]. Therefore, to ensure adequate stimulation and to avoid stimulation carry-over effects into subsequent PET scans, the tDCS parameters for this study included 5 min of stimulation at each intensity, separated by at least 10 min. The 10 min period between stimulation also allowed for the [^15^O]water tracer to sufficiently decay before the next injection. Stimulation always started with a 30 s ramp-up to the target intensity, maintenance of that intensity for 5 min, and then a 30 s ramp-down to 0 mA. The sham condition included a 30 s ramp-up to 2 mA and 30 s ramp-down at the beginning, and an extra 30 s was added at the end of this condition before PET imaging to match the total time of the other active tDCS conditions. The anode (5 cm × 7 cm) was either placed over the left DLPFC (F3 using the 10–20 convention; Experiment 1) or the left M1 (C3, adjusted to 45° from the sagittal plane [30]; Experiment 2), and the cathode (5 cm × 7 cm) was over the contralateral supraorbital area in both experiments. The subjects were asked to report any sensations experienced during stimulation and to quantify the severity of these sensations on a 10-point Likert scale (1 = “barely perceptible”, 10 = “most I could possibly stand”) [31]. The subjects were also asked to guess which stimulation condition they had experienced (sham, 1 mA, 2 mA, 3 mA, 4 mA). Feedback about the accuracy of their guesses was not provided until the subject had fully completed the study.

The injection of the [^15^O]water tracer (15 mCi/dose), and the start of the imaging, coincided with the immediate end of the stimulation ramp-down (Experiment 1) or 1 min post ramp-down (Experiment 2). Head motion was corrected for using the FUSION tool of the PMOD Biomedical Image Quantification software package (PMOD Technologies, Ltd., Zürich, Switzerland), with the first frame set as the reference. For each scan, a summed image of the 40 s immediate post-bolus transit was generated. The summed images were co-registered with the subject’s T1-weighted magnetic resonance image (MRI). Anatomically-based regions (Hammers N30R83 atlas) were determined automatically on the MRI for each individual using the PNEURO tool PMOD. Mean global activity (gCBF) was calculated based on the volume-weighted average of all intracerebral regions. The CBF relative to global was calculated for each region (rCBF = regional activity/gCBF; e.g., a ratio of 1.2 = rCBF 20% higher than gCBF). At each intensity, changes in gCBF and rCBF were investigated. Image analysis procedures were identical between the experiments, and only the focus of the rCBF regions (areas under the electrodes) was different. The mean rCBF in the same conditions from the two blocks (e.g., Block 1 sham and Block 2 sham) was calculated, and the percent rCBF change from baseline (%ΔrCBF) was determined using the following formula: %ΔrCBF = (Condition-Baseline)/Baseline). %ΔrCBF was referenced to baseline instead of sham to ascertain the presence of placebo effects from the sham protocol [32]. Additionally, to determine the consistency of the measures between the blocks, Pearson’s r for each imaging condition was calculated for each subject using the rCBF of the areas under the electrodes. Visual inspection of individual %ΔrCBF responses was performed to assess if changes were outside of the inherent [^15^O]water-PET variability (i.e., ~4% [25]). Thus, changes in rCBF ≥ ± 4% in areas under the anode and cathode were noted and interpreted based on expected (anode = increase, cathode = decrease) or unexpected (anode = decreased, cathode = increased) changes [33].

## 3. Results

### 3.1. Experiment 1, Dorsolateral Prefrontal Cortex

Individual rCBF responses for the areas under the electrodes from the different tDCS intensities are presented in Figure 2. Despite moderate–high consistency between the blocks (Pearson’s r = 0.51–0.97), %ΔrCBF was small and only two regions in subject DLPFC 01 exceeded the ≥ ± 4% threshold, both in the expected direction (−4.9% and −5.8% under the cathode in the 2 mA condition) and one region in subject DLPFC 03 in the unexpected direction (4.2% under the cathode in 1 mA). No other changes in any other condition or subject were observed.

### 3.2. Experiment 2, Primary Motor Cortex

Individual rCBF responses for the areas under the electrodes from the different tDCS intensities are presented in Figure 3. Subject Motor Cortex 01 had low consistency between blocks in the 2 mA condition (Pearson’s r = 0.21), and his one anodal increase and two cathodal decrease (%ΔrCBF = 4.2%, −4.0%, and −6.5%, respectively) could not be reliably interpreted. All other conditions across all subjects had moderate–high consistency (Pearson’s r = 0.48–0.94). Subject Motor Cortex 01 also had one increase under the anode from 1 mA tDCS (%ΔrCBF = 4.3%), two increases under the anode from 4 mA tDCS (%ΔrCBF = 6.6% and 8.8%; Figure 4), and one decrease under the cathode from 4 mA tDCS (%ΔrCBF = −7.3%). Subject 02 had an unexpected 4.1% increase under the cathode from 1 mA tDCS and an expected −5.0% and −6.9% decrease under the cathode from 2 mA tDCS. Lastly, subject Motor Cortex 03 had an unexpected 6.4% increase under the cathode from 1 mA tDCS. No other changes were noted.

### 3.3. Tolerability and Blinding Maintenance

The most commonly reported sensations in either experiment were mild–moderate burning (Likert scale (1–10) mean ± SD; sham, = 1.0 ± 0.0, 1 mA = 1.0 ± 0.0, 2 mA = 2.8 ± 1.3, 3 mA = 3.6 ± 1.4, 4 mA = 5.3 ± 1.6), and tingling (sham = 0.0 ± 0.0, 1 mA = 1.0 ± 0.0, 2 mA = 2.3 ± 1.8, 3 mA = 4.5 ± 0.0, 4 mA = 3.8 ± 3.9). Collapsing stimulation guesses across blocks and subject, 84.3% correctly guessed sham, 33.3% correctly guessed 1 mA, 16.6% correctly guessed 2 mA, 50% correctly guessed 3 mA, and 50% correctly guessed 4 mA. Furthermore, 4/6 and 2/6 subjects incorrectly guessed sham for 1 mA in Block 1 and Block 2, respectively. Taken together, the blinding data indicate that the sham protocol may have been effective for 1 mA tDCS, but only in the first block. At all other intensities, the subjects correctly guessed that they were receiving verum stimulation, even if they were unable to reliably distinguish between the intensities.

## 4. Discussion

The purposes of this preliminary investigation were to continue to explore the mechanisms of tDCS by determining individual immediate CBF responses to DLPFC tDCS in three PwMS [22] (Experiment 1) and to present individual short-term responses to M1 tDCS in three neurologically healthy adults (Experiment 2). The most notable results indicated anticipated changes in rCBF (decreased under the cathode) in two regions of one DLPFC subject in the 2 mA condition, and anode-/cathode-expected changes in one M1 subject in the 2 mA and 4 mA conditions and cathode-only changes in another M1 subject in the 2 mA condition. There were also changes contrary to the expected direction in one DLPFC subject (cathode, 1 mA) and in two M1 subjects, both under the cathode in the 1 mA condition.

The two neurotransmitters primarily implicated in cortical excitation/inhibition are glutamate and gamma-aminobutyric acid (GABA), respectively [34]. Given the strong association between glutamate and glucose [35], and the close coupling of glucose and blood flow [36], anodal increases (and cathodal decreases) in CBF from tDCS were anticipated [33]. However, despite identical stimulation parameters between the two experiments and the results of other tDCS [^15^O]water-PET studies [29,37], and the [^15^O]water-PET dose-responses of rTMS studies [38,39], such expected changes were virtually non-existent in the DLPFC subjects and were slightly more evident in a few conditions in two of the M1 subjects. Thus, these data suggest that the cortical response to tDCS might be site-specific and may require different stimulation parameters (e.g., current intensity, electrode orientation) to appropriately excite/inhibit the different target regions. Furthermore, DLPFC and M1 might have different cortical orientations/alignments [40,41] or neuronal compositions/morphologies [42] that could also contribute to site-specificity. However, this speculation certainly warrants future study before such a definitive statement can be verified. The variable responses of the individual subjects in the present report also add to the inconsistent [16], and sometimes unanticipated [15], cortical effects of tDCS.

Similarly, these results support the high variability evident in several reviews and meta-analyses [8,9,10,11,12,13] and continue to raise questions regarding the ability of sufficient residual current to reach the cortex and reliably affect brain function. Indeed, an animal and cadaver study concluded that higher intensities (≥4 mA) might be required to adequately overcome skin/skull electrical current shunting [17]. This is partially supported by the present study, which found that the largest %ΔrCBF occurred after 4 mA tDCS over M1. This is also interesting in light of another recent finding from our lab in which only higher intensity (4 mA) stimulation elicited a response to tDCS in ~50% of the subjects [43], all of whom would have been labeled as non-responders with at the lower intensity. When considered as a whole, these data support the concept of tDCS responders and non-responders complicating stimulation outcome variability and suggest that different and/or higher intensities might be required to elicit responses in some, or more, subjects. Thus, exploration of, and controlling for, known factors that affect outcome variability (e.g., biological sex [44]) is recommended [45] to advance the field into personalized tDCS applications and enhanced efficacy.

tDCS results might also be muddled by poor blinding maintenance [46,47], and the data of this report add to the growing literature on the insufficiency of a popular sham methodology to adequately blind subjects, especially in repeated-measures designs [48]. This is an important topic to explore, because breaking blinding could introduce placebo effects into tDCS outcomes [32,49,50]. For example, Bin Dawood et al. [50] found significant improvements in an orientation discrimination task in their sham tDCS condition and verified this sham-induced effect in a second experiment (also in [50]) that explored another potential explanation for their findings. They concluded that placebo effects were a likely candidate to explain their unexpected findings.

The small sample size represents the primary study limitation and reduces the generalizability of the results. However, in keeping with common PET considerations (cost and limiting the number of subjects exposed to radiation [51]) and the lengthy time each subject was on the imaging bed (≥3 h), the lack of consistent results in either experiment did not justify recruiting additional subjects. However, if tDCS responders were identified a priori, a future neuroimaging investigation using [^15^O]water or other PET radiotracers, functional magnetic resonance imaging, or arterial spin labeling might clarify the mechanisms underpinning tDCS action and would be warranted. Additionally, the 5 min stimulation time, which was designed to elicit both adequate stimulation effects and a rapid return to baseline [5,6], and imaging immediately/1 min after tDCS cessation allowed the data to be collected in a single session. However, the short stimulation duration may not have resulted in effects large enough to be detected by [^15^O]water-PET, and waiting a longer time for the effects to maximize before imaging (e.g., 10–30 [52]) might yield different results. Lastly, the age difference between the two experimental groups may have also contributed to the observed differences.

Future studies should continue to explore the mechanistic effects of tDCS on brain activity using different stimulation sites, longer stimulation durations, and longer post-stimulation imaging times. In addition, studies that combine cognitive or functional tDCS outcomes with neuroimaging, to better determine their association with brain activity, would also be of great interest. Furthermore, identification of responders and non-responders and their associated characteristics (e.g., sex, age, disease status) would be highly beneficial to advancing the tDCS field. Lastly, novel or improved sham methodology [53] and or sensation diminutions (i.e., via topical analgesics [54]) to improve subject blinding are also needed.

## 5. Conclusions

An individualized investigation of 5 min of DLPFC tDCS at sham, 1 mA, 2 mA, 3 mA, or 4 mA intensities resulted in few alterations in CBF in PwMS. Changing the anodal site to M1 elicited larger (especially in the 4 mA condition) and more CBF changes (i.e., in more subjects), suggesting the effects of tDCS might be site-specific. Still, this study highlights the high variability and individualized responses increasingly reported in the tDCS literature. In addition, the popular sham methodology used in this study may have only provided adequate blinding for the 1 mA condition in the first stimulation block, but not at other the intensities or 1 mA in the second block. Future studies should use longer stimulation durations and image at various time points after stimulation cessation when exploring the effects of tDCS on CBF. Furthermore, more studies assessing the efficacy of modified or novel sham tDCS applications to enhance subject blinding, and hence reduce placebo effects, is needed.

## Figures and Tables

**Figure 1 brainsci-10-00855-f001:**
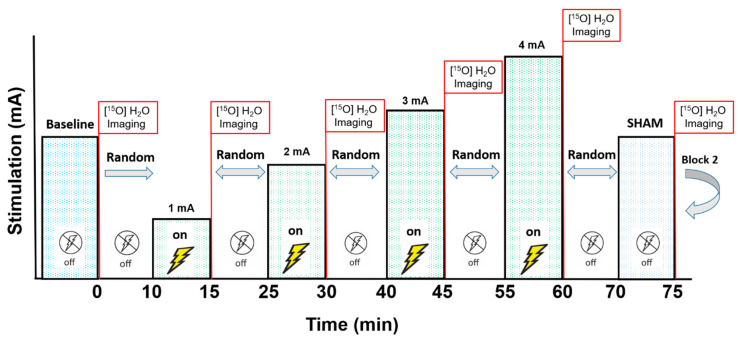
The stimulation and scan protocol. After the baseline scan, the five stimulation intensities (1 mA, 2 mA, 3 mA, 4 mA, and sham) were randomly delivered for 5 min each (six total scans per block). Ten minutes after the last scan of Block 1, Block 2 began with a new baseline scan and a different intensity randomization. Each subject experienced 12 total scans.

**Figure 2 brainsci-10-00855-f002:**
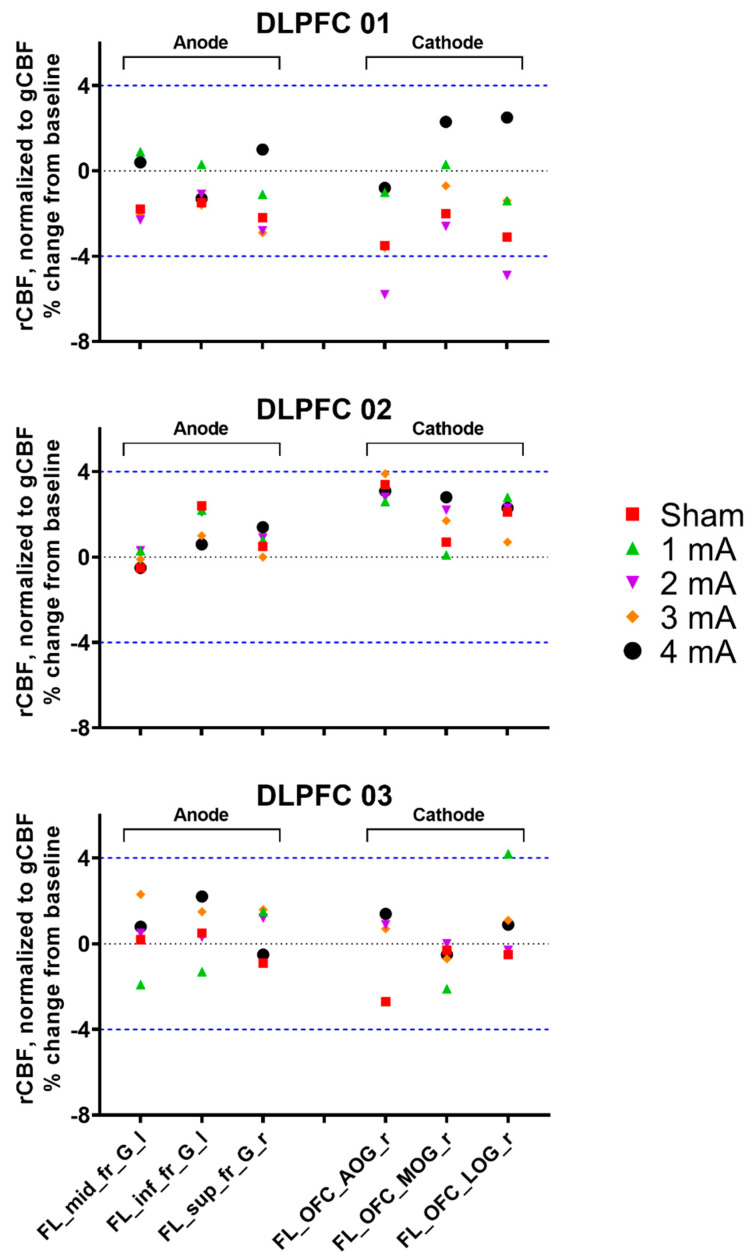
Percent change from baseline in tracer uptake (rCBF) relative to global tracer uptake (gCBF) for the three dorsolateral prefrontal cortex (DLPFC) subjects, stratified by region (areas under the electrodes) and stimulation intensity. The mean rCBF in the same conditions from the two blocks (e.g., Block 1 sham and Block 2 sham) was calculated, and the percent rCBF change from baseline was determined: %Change from Baseline = (Condition-Baseline)/Baseline). The blue dashed line represents the ≥ ±4% threshold for meaningful blood flow changes [25]. rCBF = regional cerebral blood flow; gCBF = global cerebral blood flow; FL_mid_fr_G_l = left middle frontal gyrus, FL_inf_fr_G_l = left inferior frontal gyrus, FL_sup_fr_G_l = left superior frontal gyrus, FL_OFC_AOG_r = right anterior orbital gyrus, FL_OFC_MOG_r = right medial orbital gyrus, and FL_OFC_LOG_r = right lateral orbital gyrus.

**Figure 3 brainsci-10-00855-f003:**
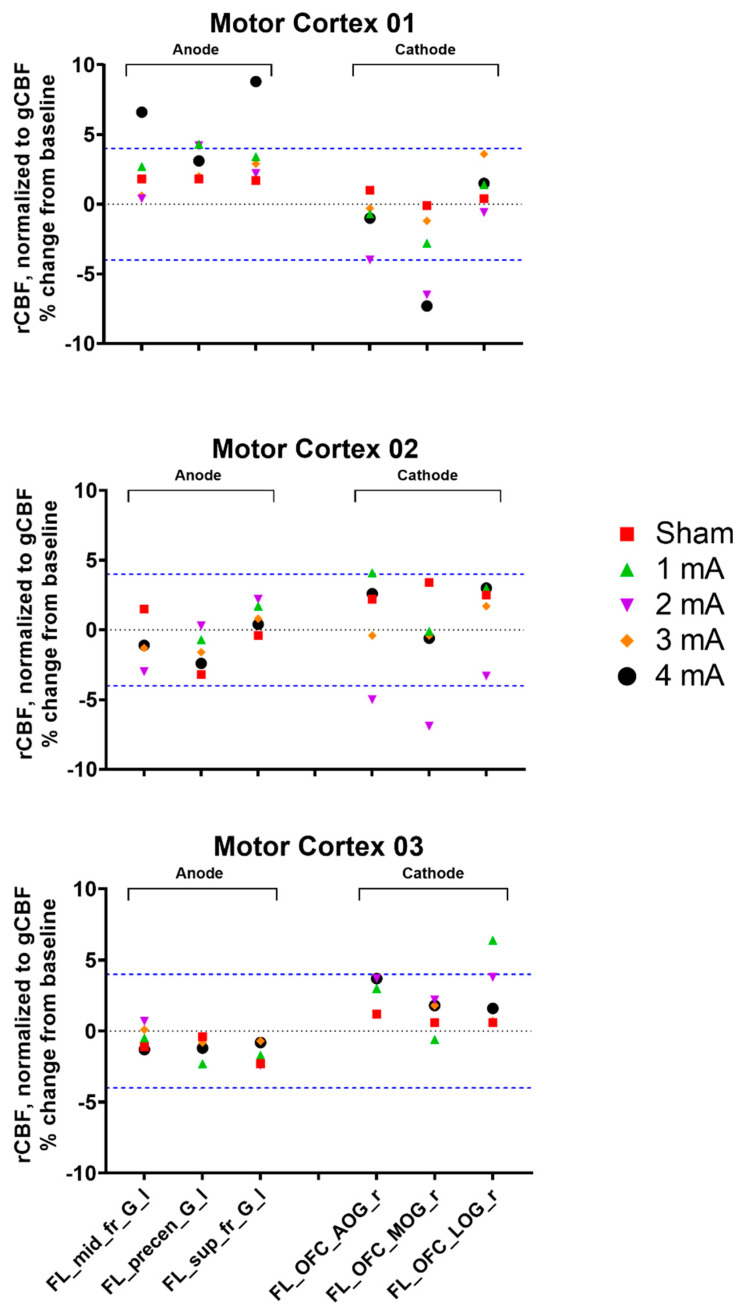
Percent change from baseline in tracer uptake (rCBF) relative to global tracer uptake (gCBF) for the three primary motor cortex subjects, stratified by region (areas under the electrodes) and stimulation intensity. The mean rCBF in the same conditions from the two blocks (e.g., Block 1 sham and Block 2 sham) was calculated, and the percent rCBF change from baseline was determined as follows: %Change from Baseline = (Condition-Baseline)/Baseline). The blue dashed line represents the ≥ ±4% threshold for meaningful blood flow changes [25]. rCBF = regional cerebral blood flow; gCBF = global cerebral blood flow; FL_mid_fr_G_l = left middle frontal gyrus, FL_precen_G_l = left precentral frontal gyrus, FL_sup_fr_G_l = left superior frontal gyrus, FL_OFC_AOG_r = right anterior orbital gyrus, FL_OFC_MOG_r = right medial orbital gyrus, and FL_OFC_LOG_r = right lateral orbital gyrus.

**Figure 4 brainsci-10-00855-f004:**
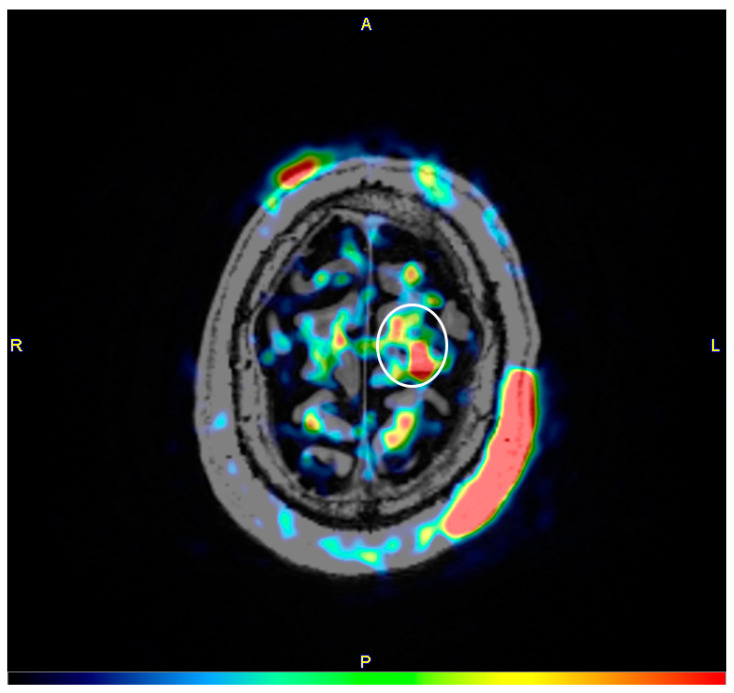
Subtraction image (4 mA tDCS–baseline) of cerebral blood flow for subject Motor Cortex 01. The image was co-registered with the same subject’s T1-weighted magnetic resonance image. Red indicates higher blood flow, followed by green and then blue. The white ellipse encapsulates brain areas with increased blood flow under the anode (left middle frontal gyrus, left precentral frontal gyrus, and left superior frontal gyrus). The large red areas on the anterior-right and posterior-left aspects of the head (top left/cathode and bottom right/anode in the figure, respectively) are increases in scalp blood circulation from tDCS (i.e., local erythema/hyperemia).

**Table 1 brainsci-10-00855-t001:** Subject demographic information (*n* = 3 in each experiment). Data are mean ± SD.

Demographic	Experiment 1	Experiment 2
Sex (M/F)	2/1	3/0
Age (years)	45.3 ± 19.0	61.0 ± 14.0
Height (cm)	171.9 ± 18.7	177.8 ± 2.5
Weight (kg)	83.5 ± 19.9	92.4 ± 9.1
Time since MS diagnosis (years)	8.0 ± 5.3	n/a
Patient-Determined Disease Steps *	2.3 ± 2.1	n/a

MS = multiple sclerosis. * Provides an indication of MS disease severity. A score of 2–4 out of 8 indicates moderate disability.

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
