# Peer review of "Individual Cerebral Blood Flow Responses to Transcranial Direct Current Stimulation at Various Intensities"

_brainsci, 2020, doi:10.3390/brainsci10110855_

Round 1

Reviewer 1 Report

The paper is good in quality, and it has a capacity of attracting readers. The following suggestions may be considered to enhance the quality and clarity of the manuscript.
1. Manuscript needs revision both at the level of concepts and write up.
2. There are too many grammatical mistakes in the manuscript. It should be corrected for the reader’s better understanding.

3. Abstract is comprehensive and well written, but it needs improvements e.g. case study can be described a little more at the end of the paragraph.

4. The introduction section is brief, hence, the motivation/gap that this paper address is not clear. reformatting may solve the issue.

5. The methodology portion needs some improvements regarding clarity and further explanations regarding techniques applied. Authors may rebuild it for a better impact on readers and viewers.

6. Some more state of the art references is needed.

7. Conclusion section requires revising to have a better impact on the readers. Add 4-5 sentences to conclusion

Author Response

Reviewer 1

The paper is good in quality, and it has a capacity of attracting readers. The following suggestions may be considered to enhance the quality and clarity of the manuscript.

  1. Manuscript needs revision both at the level of concepts and write up.
    2. There are too many grammatical mistakes in the manuscript. It should be corrected for the reader’s better understanding.

The text was carefully searched for grammatical errors and changes were made throughout the document.

  1. Abstract is comprehensive and well written, but it needs improvements e.g. case study can be described a little more at the end of the paragraph.

Specific results are better detailed in the Abstract (lines 21-24).

  1. The introduction section is brief, hence, the motivation/gap that this paper address is not clear. reformatting may solve the issue.

Text was added throughout the Introduction (especially lines 77-80) to clarify the purpose and anticipated findings of the study.

  1. The methodology portion needs some improvements regarding clarity and further explanations regarding techniques applied. Authors may rebuild it for a better impact on readers and viewers.

Several alterations and clarifications to the Methods (especially lines 116-120) were made throughout this section.

  1. Some more state of the art references is needed.

We strive to have up-to-date references whenever possible and most of our citations are from 2014 or later. However, certain publications are foundational to the field, as is the case with the older citations for tDCS (e.g., Nitsche and Paulus 2000, 2001) or for [15O]water-PET (e.g., Herscovitch et al., 1983; Raichle et al., 1983) and do not have any updates.

  1. Conclusion section requires revising to have a better impact on the readers. Add 4-5 sentences to conclusion

Additional content was added to the Conclusion (lines 278-288).

Reviewer 2 Report

In this study, the authors examined the cerebral blood flow response at two locations to different intensities of tDCS. The aims of the study were to examine changes in CBF at an individual level and to evaluate intra- and inter-individual differences in response to tDCS. The study also undertook to examine the effects of tDCS in two different areas and in clinical and control cohorts. The authors demonstrate high variability and individualized responses across intensities of tDCS studied and locations stimulated.

One of the strengths of the study is that it is using PET to measure cerebral blood flow changes caused by different intensities of tDCS. However the sample size in this study is small and with no real statistical analysis of data including no correction for multiple comparisons. In addition, the following weaknesses in the manuscript need to be addressed by the authors.

Introduction: It is not clear what the study hypothesis is. For instance, why is DLPFC the target for anodal tCDS in individuals with MS? What MS symptoms are targeted by stimulating DLPFC either with anodal tDCS? While indeed this technique has several desirable characteristics, including ease of use, low cost, and possibilities for in-home use, the most important use should be directed towards a specific symptom of MS. Are the tDCS induced responses related to inhibitory or excitatory thresholds?

Methods: Many methodological details are lacking. What were the electrode dimensions? There is no description of how sham tDCS administered. What was the dose of [15O]water injected? How was head motion during PET imaging corrected? Task performance can also add variability in performance and habituation. How did the authors confirm that the participants were performing the task accurately. The motor/cognitive task might have saturated the CBF increase in the ROIs so that any additional (likely smaller) tDCS induced changes are not evident. How were the ROIs defined in each individual? Did the ROIs had fixed volume or were variable in each individual? Were ROI thresholded by anatomical margins or t or z scores?

Results: Using ³4% change from baseline to be significant appears to an arbitrary cutoff. Why not use, statistical significance instead? There is no correction for multiple comparisons. There appears to be age difference between two groups which may result in differential tDCS responses.

Discussion: The authors may want to draw comparisons to TMS intensity studies in M1 and DLPFC (Speer et al., 2003 and Fox et al., 2006). Inhibitory interneurons have lower threshold than excitatory neurons. The authors should discuss whether the response to tDCS is related to primarily to intensity of stimulation, irrespective of anode or cathode stimulation.

References:

  1. Speer, Andrew M, Mark W Willis, Peter Herscovitch, Margaret Daube-Witherspoon, Jennifer Repella Shelton, Brenda E Benson, Robert M Post, and Eric M Wassermann. “Intensity-Dependent Regional Cerebral Blood Flow during 1-Hz Repetitive Transcranial Magnetic Stimulation (RTMS) in Healthy Volunteers Studied with H215O Positron Emission Tomography: I. Effects of Primary Motor Cortex RTMS.” BIOL PSYCHIATRY, n.d., 8.
  2. Speer, Andrew M, Mark W Willis, Peter Herscovitch, Margaret Daube-Witherspoon, Jennifer Repella Shelton, Brenda E Benson, Robert M Post, and Eric M Wassermann. “Intensity-Dependent Regional Cerebral Blood Flow during 1-Hz Repetitive Transcranial Magnetic Stimulation (RTMS) in Healthy Volunteers Studied with H215o Positron Emission Tomography: II. Effects of Prefrontal Cortex RTMS.” Biological Psychiatry 54, no. 8 (October 2003): 826–32.
  3. Fox, Peter T., Shalini Narayana, Nitin Tandon, Sarabeth P. Fox, Hugo Sandoval, Peter Kochunov, Charles Capaday, and Jack L. Lancaster. “Intensity Modulation of TMS-Induced Cortical Excitation: Primary Motor Cortex.” Human Brain Mapping 27, no. 6 (June 2006): 478–87.

Author Response

Reviewer 2

Introduction: It is not clear what the study hypothesis is. For instance, why is DLPFC the target for anodal tCDS in individuals with MS? What MS symptoms are targeted by stimulating DLPFC either with anodal tDCS? While indeed this technique has several desirable characteristics, including ease of use, low cost, and possibilities for in-home use, the most important use should be directed towards a specific symptom of MS. Are the tDCS induced responses related to inhibitory or excitatory thresholds?

This investigation was more mechanistic (direct stimulation effects of tDCS on the brain) than it was functional. Our choice of stimulation site for Experiment 1 (which was previously published as a group-wise investigation; see citation [22]) was DLPFC. As we mentioned in this previous publication, this site was chosen to avoid potentially confounding brain activity from the counting task and spontaneous movement that may have occurred during the scanning. We clarified this choice in the last paragraph of the Introduction (lines 70-71). We also more clearly state our expectations/hypotheses at the end of this same paragraph (lines 77-80).

Methods: Many methodological details are lacking. What were the electrode dimensions? There is no description of how sham tDCS administered. What was the dose of [15O]water injected? How was head motion during PET imaging corrected? Task performance can also add variability in performance and habituation. How did the authors confirm that the participants were performing the task accurately. The motor/cognitive task might have saturated the CBF increase in the ROIs so that any additional (likely smaller) tDCS induced changes are not evident. How were the ROIs defined in each individual? Did the ROIs had fixed volume or were variable in each individual? Were ROI thresholded by anatomical margins or t or z scores?

The dimensions of the electrodes were added (line 120 and 122), the tDCS procedures were detailed (lines 116-120), the tracer dose was added (line 128), motion correction was explained (lines 130-132), in-scan task performance monitoring was detailed (lines 92-93), and ROI definition was further clarified (lines 134-135).

Results: Using ³4% change from baseline to be significant appears to an arbitrary cutoff. Why not use, statistical significance instead? There is no correction for multiple comparisons. There appears to be age difference between two groups which may result in differential tDCS responses.

Regarding statistics, because our primary purpose for this study was to describe individual responses (and not group responses), the only statistic calculated was the between-block consistency analysis (Pearson’s r) and we did not perform any significance testing. Importantly, we intentionally avoided using the word “significant” when presenting and discussing our results for this paper, precisely to avoid conflating our observations with statistical significance. Instead, we use words like “noted” and “observed.” Additionally, as mentioned in the manuscript (lines 146-150), the 4% threshold is based off of a [15O]water-PET study (Herscovitch et al. 1983) in which they found the differences between PET blood flow and directly measured blood flow were ~4%. Using this threshold helped ensure our changes were outside of the noise of the [15O]water-PET measurement. Lastly, we agree that the age difference may have influenced the findings and added this as a limitation in the Discussion (lines 267-268).

Discussion: The authors may want to draw comparisons to TMS intensity studies in M1 and DLPFC (Speer et al., 2003 and Fox et al., 2006). Inhibitory interneurons have lower threshold than excitatory neurons. The authors should discuss whether the response to tDCS is related primarily to intensity of stimulation, irrespective of anode or cathode stimulation.

References:

  1. Speer, Andrew M, Mark W Willis, Peter Herscovitch, Margaret Daube-Witherspoon, Jennifer Repella Shelton, Brenda E Benson, Robert M Post, and Eric M Wassermann. “Intensity-Dependent Regional Cerebral Blood Flow during 1-Hz Repetitive Transcranial Magnetic Stimulation (RTMS) in Healthy Volunteers Studied with H215O Positron Emission Tomography: I. Effects of Primary Motor Cortex RTMS.” BIOL PSYCHIATRY, n.d., 8.
  2. Fox, Peter T., Shalini Narayana, Nitin Tandon, Sarabeth P. Fox, Hugo Sandoval, Peter Kochunov, Charles Capaday, and Jack L. Lancaster. “Intensity Modulation of TMS-Induced Cortical Excitation: Primary Motor Cortex.” Human Brain Mapping 27, no. 6 (June 2006): 478–87.

Thank you for providing these interesting rTMS studies. We have added a phrase discussing their intensity dose-response to the Discussion (line 225). Regarding inhibitory vs. excitatory neuronal thresholds, the tDCS verbiage of “excitation” and “inhibition” doesn’t match with “excitatory neurons” and “inhibitory neurons” per se. Indeed, unlike TMS, the neuronal effects of tDCS are entirely subthreshold and are only purported to alter the resting potential of a given neuron; anodal tDCS slightly depolarizes the resting potential and cathodal tDCS slightly hyperpolarizes the resting potential. These depolarization and hyperpolarization effects are also dependent on neuronal orientation and morphology (see lines 229-231 of the present manuscript for relevant citations) and are likely independent of neuronal function (excitatory vs. inhibitory).

Round 2

Reviewer 2 Report

The authors have sufficiently addressed my concerns.